# Prostate-Specific Antigen Monitoring Using Nano Zinc(II) Metal–Organic Framework-Based Optical Biosensor

**DOI:** 10.3390/bios12110931

**Published:** 2022-10-27

**Authors:** Said M. El-Sheikh, Sheta M. Sheta, Salem R. Salem, Mohkles M. Abd-Elzaher, Amal S. Basaleh, Ammar A. Labib

**Affiliations:** 1Department of Nanomaterials and Nanotechnology, Central Metallurgical R & D Institute, Cairo 11421, Egypt; 2Department of Inorganic Chemistry, National Research Centre, Cairo 12622, Egypt; 3Department of Biochemistry, Egypt Centre for Research and Regenerative Medicine, Cairo 11887, Egypt; 4Department of Chemistry, Faculty of Science, King Abdulaziz University, P.O. Box 80203, Jeddah 21589, Saudi Arabia

**Keywords:** prostate antigen, photoluminescence, nanoparticles, metal–organic framework based on zinc(ii), biosensor

## Abstract

Background: The prostate-specific antigen (PSA) is an important cancer biomarker that is commonly utilized in the diagnosis of prostate cancer. The development of a PSA determination technique that is rapid, simple, and inexpensive, in addition to highly accurate, sensitive, and selective, remains a formidable obstacle. Methods: In this study, we developed a practical biosensor based on Zn(II) metal–organic framework nanoparticles (Zn-MOFs-NPs). Many spectroscopic and microanalytical tools are used to determine the structure, morphology, and physicochemical properties of the prepared MOF. Results: According to the results, Zn-MOFs-NPs are sensitive to PSA, selective to an extremely greater extent, and stable in terms of chemical composition. Furthermore, the Zn-MOFs-NPs did not exhibit any interferences from other common analytes that might cause interference. The detection limit for PSA was calculated and was 0.145 fg/mL throughout a wide linear concentration range (0.1 fg/mL–20 pg/mL). Conclusions: Zn-MOFs-NPs were successfully used as a growing biosensor for the monitoring and measurement of PSA in biological real samples.

## 1. Introduction

In males, the prostate gland is a common site for the development of prostate cancer, one of the most rapidly progressing forms of malignant tumors [1,2,3,4]. It is the second highest cause of cancer patient fatalities in the United States, and constitutes 7.2% of the total number of cancer cases in Egypt, as estimated by Global Cancer Observatory (GLOBOCAN) in December 2020, which makes it the fifth most prevalent cancer [5,6,7]. In the vast majority of prostate cancer cases, symptoms do not manifest until later stages [8]. It is preferable to diagnose this sort of tumor as early as possible to reduce the mortality rate using normal therapy procedures [9]. Recent statistics from the World Health Organization (WHO) indicate an estimated prostate cancer death rate of around 1.7 million per year, with a rise of approximately 0.5 million by 2030 [6,10].

PSA, or prostate-specific antigen, is a glycoprotein made up of 237 amino acids that is released by the prostate gland [11]. It is used as an important biomarker for diagnosing prostate cancer by measuring its concentration in blood samples [11]. Serum PSA concentrations between 4.0 and 10.0 ng/mL are considered normal; a concentration of PSA more than 10.0 ng/mL is indicative of a carcinoma of the prostate in its first stages [12,13].

Owing to the widespread prevalence of malignant neoplasm of the prostate in men, several scientific and medical research investigations have focused on the development of simple, rapid, affordable, and accurate PSA-based-methods for early prostate cancer diagnosis, including electrochemical and fluorometric methods [14,15,16,17,18,19,20,21,22,23,24,25,26]. Unfortunately, the systems and procedures described above offer several benefits and drawbacks. Each method or technique has advantages such as sensitivity and accuracy, but is intensively solvent using and time-consuming, etc. The development of technologies and novel approaches for the detection of PSA is of major importance for early diagnosis. Accordingly, Karami et al. [27] and Yazdani et al. [28] summarized all methods that can be used in PSA detection by detecting nano-elements and their characteristics such as detecting element linear range and limit detection. Additionally, they study the parameters influencing the performance of the sensor to achieve high sensitivity [18,27,28]. Many review articles summarized different electrochemical aptamer-based biosensors for the detection of PSA [29]. They emphasized the specific performance parameters, advantages, and disadvantages of different methods, nanomaterials, and kits developed to attain highly selective and sensitive for the determination of PSA [29]. Moreover, Jalalvand et al. [30] reported many analytical techniques used for PSA detection and sometimes combined two methods such as electrochemical with chemometric methods to enhance the performance. They also mentioned several advantages of electrochemical biosensors such as good selectivity and sensitivity which has received wide attention from researchers [8,31,32]. However, most of these techniques did not have selectivity and sensitivity enough for PSA detection [33]. Herein, the fluorescence-based methods in comparison with traditional detection methods are more sensitive and faster [34].

On the other hand, the current challenges and prospects of nanomaterials-based PSA detection methods are essential. A few reports about using MOFs for PSA detection were published. Zhang et al. [35] reported an electrochemical immunosensor using MOF-235 for the sensitive detection of PSA [35]. The common nanomaterials-based biosensors used for prostate cancer (PSA) detection were reviewed and emphasized [33,36,37].

Therefore, in this paper, we aim to build a simple and rapid technique that combines the benefits of the aforementioned approaches and reduces their limits by leveraging the applicability, selectivity, and sensitivity of nanotechnologies [38,39,40,41,42,43,44,45,46,47]. Metal–organic frameworks nanoparticles (MOF-NPs) offer many desirable qualities, including stability both chemically and thermally, relatively high surface area and porosity, high magneticity, extremely efficient absorbency, and so on [48,49,50,51]. MOF-NPs, due to their superior physicochemical characteristics, are very selective and sensitive methods for forming bonds with both organic and biological molecules, and they may be dealt with more effectively than the rest of the components by relatively simple means [52]. In addition, MOF-NPs’ unique features allow them to be used in a wide range of fields, from sensing and biomedical to drug delivery and gas separation/storage, to catalysis and beyond [53,54,55,56,57,58].

Herein, Zn-MOFs-NP was synthesized via reacting zinc acetate with N1,N3-bis(2-(3-((l1-oxidaneyl)carbonyl)-5-aminobenzamido)phenyl)-5-aminoisophthalamide as a nano-organic linker which was previously reported [56]. Various micro/analytical techniques were employed to demonstrate that the synthesized Zn-MOFs-NP had the suggested structure and form. The PL study findings influenced the selection of Zn-MOFs-NP as a biosensor for detecting and measuring prostate-specific antigen (PSA). PSA concentration increases were shown to improve the PL emission spectrum of Zn-MOFs-NP. The MOF compound might thus be directed to use as a biosensor to quantify PSA levels. In addition, common tumor biomarkers and other biomolecules did not demonstrate any impact. The present method has also been compared to earlier research. The Zn-MOFs-NP-based PSA detection approach was superior to the others in terms of throughput, simplicity, cost, sensitivity, selectivity, and usability. Analytical and statistical analyses, as well as testing on the improvement mechanism and acute thermal stability, were performed on the proposed biosensor to see whether it could be utilized with actual blood samples [59,60].

## 2. Materials and Methods

### 2.1. Materials

Zinc acetate (99.99%) and 1, 2-phenylenediamine (99.5%) were acquired by Sigma- Aldrich. From Acros-organics 5-aminoisophthalic acid (98%) was purchased. Different buffered concentrations of standard-PSA were acquired from Monobind, USA. Both chemicals and solvents used as received in this study were analytical reagent grade.

### 2.2. Instruments

The FE-SEM/EDX spectroscopy was carried out with “FE-ESM-JEOL-JSM-6510LV- Japan”. The phases of structures were assessed by “HR-TEM-JEM-2100-JEOL, Japan”. The solid sample of a Zn-MOFs-NP was applied for mass spectroscopy using “Thermo-Scientific-mass spectrometer-USA”. FT-IR spectra in a range (400–4000 cm^−1^) were performed by “JASCO-FT-IR-460-spectrophotometer, USA” using KBr tablets. C-H-N-elemental analysis was accomplished via “ECS-4010-Costech analyzer-Italy”. UV-vis spectra were performed using “JASCO V-770 UV-Visible/NIR spectrophotometer-USA” and the software used for calculation of the energy band gap was Optbandgap-204B. The Zn-MOFs-NP species and oxidation states were verified by “Thermo Scientific X-ray photoelectron spectrometer, USA”. The thermogravimetric and thermal behavior analysis of the Zn-MOFs-NP (DSC/TGA) was performed by “Universal V4.5-TA analyzer-USA”. The Zn-MOFs-NP magnetic futures were investigated by “7400-1-VSM magnetometer-USA”. The photoluminescence (PL) investigation was carried out with “Shimadzu-RF-5301PC-spectrofluorophotometer-Japan). The samples measurements were accomplished in a 1.0 cm quartz cuvette path length at 30 s scan time and at 25 °C. The software used for data analysis was Origin-8, whereas the program used for drawing the geometrical 3D structures and Schemes was “ChemBioDraw-Ultra12”.

### 2.3. Synthesis of Zn-MOFs-NP

The MOF nanoparticle of Zn(II) was produced by adding 2.0 mmol of zinc II acetate dropwise to the synthesized *N1*,*N3*-bis(2-(3-((l1-oxidaneyl)carbonyl)-5-aminobenzamido)phenyl)-5-aminoisophthalamide (ABAPAPA) linker which was prepared according to previous report [56], followed by 24 h of stirring and reflux at 80 °C. After a time, the solution became brown, and an off-white precipitate formed, which was filtered, washed, and eventually dried at room temperature (Appendix A).

### 2.4. PL-Measurements Procedures

By dissolving an adequate quantity of Zn-MOFs-NP in DMSO, a concentration of (1 mM) stock solution was obtained. A working solution (10 mM) was produced by diluting the stock solution with deionized water. Then, the sample solution was evaluated for PL measurements to determine the optimal excitation wavelength by choosing the maximum wavelength emission. The PL spectrum of Zn-MOFs-NP (10 mM) was investigated against appropriate concentrations that were freshly prepared of PSA, as biomolecules and significant tumor biomarkers. The PL-intensity at ƛ_Em_ = 366 nm after an excitation wavelength of 305 nm was shown to be linearly related to PSA concentration between 0.1 and 0.2 fg/mL under the most ideal PL measuring circumstances. Parameters of the equation, including the correlation coefficient, were calculated using the least-squares method of statistics. The equation that was used was (Y = n + n’X), where Y represents the PL intensity of the Zn-MOFs-NP at em = 372 nm, n represents the intercept, n’ represents the slope, and X is the concentration of PSA. Moreover, the LOQ and LOD were determined using the following equations: ($LOQ=10\;\left(\frac{S}{b}\right)$ & $LOD=3.3\;\left(\frac{S}{b}\right)$) [61,62,63]. (*S*) is the standard-error value of the PL intensity; (*b*) is the slope of the linear-graph. Zn-MOFs-NP PL spectra were also examined to those of true blood samples that had been treated with various amounts of PSA.

### 2.5. PSA Determination by MOF in Real-Samples

The serum samples that are considered to be representative were taken from a disposable sample that was located in a medical laboratory. The samples were evaluated and processed in accordance with the standard recommendations and safety procedures.

## 3. Results

The Zn-MOFs-NPs were made by reacting zinc II acetate with the *N1*,*N3*-bis(2-(3-((l1-oxidaneyl)carbonyl)-5-aminobenzamido)phenyl)-5-aminoisophthalamide compound in nano form represented as a linker in a simple way, as shown in Appendix A (Reaction Scheme). After obtaining a precipitate with a somewhat off-white color, the substance was filtered, then washed, and, lastly, dried. Based on the collected micro/analytical data, a suggested structure was performed and evaluated.

### 3.1. FE-SEM, EDX and HR-TEM

The FE-SEM and EDX images of Zn-MOFs-NP are presented in (Figure 1a–c). The FE-SEM images of Zn-MOFs-NP at different magnifications (Figure 1a,b) looked to be an aggregation of inconsistent square forms (1a) and nearly inconsistent square wood shapes with proportionally sized dimensions of around 118 nm (1b). The mapping analysis of the Zn-MOFs-NPs using EDX (Figure 1c and Table 1) revealed the existence of carbon, oxygen, nitrogen, and zinc as a building block element in every single particle surface. EDX mapping (Figure 1c) confirmed the formation of the Zn-MOFs-NP structure by displaying a uniform distribution of block components throughout the cross-section. Similarly, the mapping element percentages from the EDX Table (Table 1) were in close agreement with the approximate proportion of theoretical elements. Theoretically, C is 40.18, N is 6.83, O is 28.99, and Zn is 18.23. In practice, C is 40.00, N is 6.57, O is 34.87, and Zn is 18.56. The Zn-MOFs-NP TEM picture (Figure 1d) shows irregular square nanosheets with a moderate size of around 120 nm. The SAED and high magnification pictures of the 3D nanostructure of square sheets are shown in (Figure 1e,f). High-resolution TEM (1e) and its selected area electron diffraction (SAED) pattern (1f) suggest the high crystallinity of Zn-MOF, which was confirmed by XRD.

### 3.2. Elemental Analysis

Table 1 displays the EDX mapped and theoretically predicted CHN-elemental data for the Zn-MOFs-NP compound. The findings were in good agreement with the predicted molecular formulas; the Anal./Calc. (percent): C_48_H_82_N_7_O_26_Zn_4_, (1434.730 gmol^−1^), N, 6.83; H, 5.76; C, 40.18; found N, 6.82; H, 5.95; C, 39.96; with a reaction yield of around 87.80 percent.

### 3.3. Mass Spectrum

The mass spectrum of the Zn-MOFs-NP compound (Appendix A) and the proposed fragmentation strategy (Appendix A) were illustrated. The figures depicted the *m*/*z* peaks that were perfectly consistent with the suggested empirical formula (C_48_H_82_N_7_O_26_Zn_4_), as computed theoretically and validated by C\H\N-analysis, a molecular ion-peak was observed at 1434.730 *m/z* for the Zn-MOFs-NP compound. Following fragmentation of the ion with *m*/*z* = 1434.730, Appendix A also displays several significant peaks at *m*/*z* = 943.0, 716.0, 488.0, 229.0, 171.0, 149.0, 108.0, and 65.0 caused by the loss of (ethanol and water) molecules and the subsequent breakdown of the organic structure. The Zn-MOFs-NP compound fractured in most cases in accordance with the mass spectrum, computed theoretical fragmentations, and presumed structures of molecules.

### 3.4. UV-Vis and FT-IR Spectra

In contrast to the nano organic linker, the spectra chart of the IR spectroscopy of Zn-MOFs-NP is displayed in Appendix A. The typical intense peaks at 3440.0, 3140.0, 3018.0, 2928.0, and 1157.0 cm^−1^ are caused by the stretching band of the compound’s NH_2_ groups, H_2_O, and Et-OH molecules [64]. Continuing the interpretation, the peak at 3263.0 cm^−1^ is attributed to N-H stretching. It was found that the presence of sharp bands at 1640.0, 1585.0, 1560.0, 1476.0, and 1364.0 cm^−1^ also are attributed to C=O, C=N, and C=C stretching [41,65]. CH is assigned to the bands between 1024.0 and 771.0 cm^−1^. For the transition metal ions, bands at 540.0 and 418.0 cm^−1^ correspond to a zinc ion that interacted with O and N either with the formation of coordination and covalent bonds v(Zn−O) and (Zn−N), respectively [65,66,67]. The emergence of the new intense peaks above verified the formation of coordination and covalent sharp bond of the zinc ion and the nano *N1,N3*-bis(2-(3-((l1-oxidaneyl)carbonyl)-5-aminobenzamido)phenyl)-5-aminoisophthalamide compound linker through the N and O. The reflection spectra (Appendix A) and bandgap energies (BGE) (Appendix A) were studied for Zn-MOF with nano-organic linkers. The Zn-MOF compound exhibited diverse reflection bands at 235.0, 271.0, 330.0, 385.0, 408.0, and 634.0 nm, as shown in Appendix A. These bands might be caused by LMCT and intra-ligand -*, n-* [68]. The BGE spectra in Appendix A further show that the Zn-MOFs-NP has lower BGE values than the linker at 1.79, 2.95, and 3.50 eV due to conjugation inside the *N1,N3*-bis(2-(3-((l1-oxidaneyl)carbonyl)-5-aminobenzamido)phenyl)-5-aminoisophthalamide skeleton, which results in a higher BEG for the HOMO valance [41,55].

### 3.5. XRD and XPS Analysis

The XRD chart of different Zn-MOFs-NP compounds was presented in (Figure 2a). The XRD patterns of the Zn-MOFs-NP showed sharp strong peaks, indicating that the high crystalline-phase of the Zn-MOFs-NP had developed. Moreover, the diffraction patterns corresponded to normal Zn-MOF and other produced Zn-MOFs-NP XRD patterns, demonstrating the effectiveness of the Zn-MOFs-NP Synthesis method [69,70]. The nano crystallite size was estimated by the Scherrer equation provided in Appendix A, which showed the crystallite size was in the range of 120 nm. This finding has corroborated the SEM and TEM findings.

The XPS spectra (Figure 2b–e) revealed the existence of C, O, N, and Zn, only with no traces of contamination. The existence of Zn^2+^ in the Zn-MOFs-NP was confirmed by the detection of a signal at 1020.92 eV in the XPS spectra of Zn 2p, which was related to the Zn(II) 2p3/2 satellite peak (Figure 2b) [71]. The O 1s spectra revealed three peaks of O-Zn-O at 530.34 eV, C-O at 531.25 eV and C=O at 533.22 eV. While the satellite peak in the N 1s spectra was 398.16 eV, concurrently, the C 1s spectrum revealed the presence of three signals at 282.85 eV (C-C), 286.52 eV (C-N) and 289.59 eV C=O (Figure 2b–e) [71].

### 3.6. Thermal Behavior and Stability of the Zn-MOFs-NP

TGA/DSC plots of Zn-MOFs-NP indicated that the Zn-MOFs-NP compound broke down in four phases (Figure 2f). At temperatures ranging from 65 to 226.0 °C, the initial and subsequent weight losses are attributable to the liberation of C_2_H_5_OH and intra/inter H_2_O molecules. The organic skeleton began to degrade around 440.0 °C. This resulted in the third and fourth breakdown phases. The remainder is zinc (18.11 percent). The discussed data agreed with the results of mass spectrometry and XRD. The thermogravimetric behavior of the Zn-MOFs-NP further suggests that at temperatures up to 400.0 °C the Zn-MOFs-NP is thermally stable [72,73].

The whole presented data and discussion may be used to determine the structure of prepared Zn-MOFs-NP that exists in three dimensions and its exceptional molecular surface (Figure 3a,b). Accordingly, the 3D structure of Zn-MOF plates and morphology have intrinsic properties such as photonic bandgap, microporosity, high-spatial density, low surface energy, the defects and edge positions, complexity over length, and scales ranging and integrated with functional low-dimensional building blocks which can enhance the performance of sensing properties. Geometry with special crystal facets can provide a high large surface area and active sites, which is critically significant for highly efficient sensors.

### 3.7. Photoluminescence Investigation

Specifically, the PL curve was made via graphing the data of the excitation spectrum versus the emission spectra of Zn-MOFs-NP (Figure 4a). Zn-MOFs-NP photoluminescence (PL) spectra were measured and plotted (at different excitation wavelengths) in Appendix A. At an excitation wavelength of 305 nm (10 mM), the PL spectrum of Zn-MOFs-NP revealed a rise of emission band located at 366 nm. Furthermore, Zn-MOFs-NPs may be employed as optical biosensors for PSA detection and measurement.

ZMOF-NPs photoluminescence spectrum was analyzed in relation to a range of prostate-specific antigen concentrations (Figure 4b). It was found that the increase in the PSA concentrations significantly enhanced the Zn-MOFs-NP PL peak intensity in a PSA range of concentrations (0.1–0.2 fg/mL) with a small red-shift (about six nm) from 366.0 to 372.0 nm. After taking into consideration technique evaluations and statistical characteristics, the results demonstrated that ZMOF-NPs has the potential to be deemed a suitable biosensor based on spectrofluorimetric phenomena for the detection and quantification of PSA.

### 3.8. Method Validation

The calibration curve demonstrates the relationship between the PL intensity of ZMOFNP measured at ƛ_Em_ = 372.0 nm and the concentration of PSA in the sample, which may give rise in between 0.1 and 0.2 fg/mL (Appendix A). The linear-dynamic relationship for the Zn-MOFs-NP PL biosensor was observed (Figure 4c) under optimal circumstances for PL measurements. Rising PSA concentration significantly increased the spectrum PL intensities (Figure 4c). Over the range of interest (0.1–1000.0 fg/mL), the calibration curve was graphed linearly.

Zn-MOFs-NP PL-intensity = 105.21 log [PSA] + 523.01 with r^2^ = 0.983.

The limit of detection and quantification (LOD and LOQ) of the optical photoluminescence biosensor based on ZN-MOFS-NP was calculated to be 0.145 and 0.438 fg/mL, respectively, where Appendix A summarizes the results of the analysis of data using PL regression. The suggested optical photoluminescence biosensor’s low LOD and LOQ values and large linear ranges of concentrations are validated by table data. Furthermore, (Table 2) compares the new biosensor’s performance to other prior studies for PSA measurement and determination, where the present optical photoluminescence biosensor has a lower LOQ and LOD, and a broader linear detection PSA range compared to previous methods of detection.

Studies were conducted to determine the selectivity and specificity of the current optical photoluminescence Zn-MOFs-NP-based biosensor in the presence of competing analytes, such as biomolecules and other tumor biomarkers. Through the use of the PL spectrum of Zn-MOFs-NP (10.0mM) against PSA (100.0 fg/mL), CEA “carcinoembryonic antigen” (10.0 ng/mL), AFP “alpha-fetoprotein” (10.0 ng/mL), and CK-T “creatine kinase total” (10.0 ng/mL), selectivity and specificity studies were collated and exhibited in a histogram (Figure 4d). From the histogram, it can be noted that the luminescence intensity in cases of PSA is highly enhanced, whereas the other interfering analytes (similar biomarkers or organic molecules) do not make significant changes in the luminescence intensity of the Zn-MOFs-NP-based biosensor, which demonstrates the high selectivity and specificity of the proposed biosensor.

The curve demonstrates that PSA induced a considerable rise in the intensity of the Zn-MOFs-NP PL with a minor redshift, but the other interfering matrix had no reaction. As a result, we may infer that Zn-MOFs-NPs are very specific and selective for PSA when compared to other interfering matrices. Zn-MOFs-NP-based optical photoluminescence biosensors for PSA detection and measurements in genuine human blood serum samples were evaluated for their precision, accuracy, recovery, and usability. In this study, the PSA standard was spiked into blood samples at doses of 1.0, 100.0, 500.0, 1000.0, and 2000.0 fg/mL.

Three separate repetitions of each test were performed (Table 3). The mathematical calculations and statistical evaluations showed that the mean values (X) were suitable and in the range. The suggested method is precise and accurate since the observed sharp decrease in the values of relative error (RE%) averaged (1.01%) and the relative standard deviation (SD) averaged 4.270%. Importantly, the average percent recovery (RC%) was 99.02%, proving the existing optical photoluminescence biosensor is sensitive enough to detect prostate-specific antigen as an important cancer biomarker diagnostic for the prostate with pinpoint precision at very low concentrations.

### 3.9. Interaction Mechanism

The fluorescence mechanism of the enhancement process may be extrapolated from (Figure 4b), which indicates that when PSA concentrations increased, so did the PL-emission intensities of Zn-MOFs-NP. The shift in Zn-MOFs-NP emission intensities recorded by increasing PSA concentrations indicates the nature of the MOF-PSA interaction. The observed biochemical sensing behavior of Zn-MOFs-NP towards PSA might be ascribed to PSA’s remarkable affinity for the amine group lone pair of electrons inside Zn-MOFs-NP [64]. Furthermore, because the Zn-MOFs-NPs have low-lying “*orbitals” in amine groups of phenyl rings, the fluorescent performance of the Zn-MOFs-NP and further explanation that the interaction mechanism might be because of molecular-orbital transitions inside ligand-metal charge transfer (LMCT) resulted from intra-ligand n/-* transitions. Furthermore, the increased probability of Zn-MOFs-NP chelating with PSA as a biological target may be explained by the potential of “electron” delocalization inside the chelation ring and partial sharing of opposing positive charges with the donor-groups. According to chelation theory, this delocalization and charge-sharing process increases the lipophilic properties of the Zn-MOFs-NP and, as a result, decreases the polarity of the zinc-ion [52], implying that the interaction of Zn-MOFs-NP with PSA may be due to the formation of covalent bonds [65].

## 4. Conclusions

This scientific study describes an optical biosensor that employs photoluminescence and is based on a novel, promising Zn-MOFs-NP that has been thoroughly evaluated. The Zn-MOFs-NP morphology looked to be inconsistent square wood forms with an appropriate size of around 118 nm. The thermal stability measurements further demonstrated that the Zn-MOFs-NPs were considered stable. The photoluminescence investigation findings suggested that Zn-MOFs-NP might be employed as an advantageous optical biosensor for determining PSA content in actual human blood serum. From a statistical perspective, the current method served its intended function well. The provided results of the suggested approach showed lower PSA LOD with a response that was faster, easier, and cheaper than those reported in the literature. Moreover, the data unveiled in the not-so-distant future will be an essential piece for monitoring and quantifying PSA, a useful diagnostic tool for identifying prostate cancer early on—which is the most frequent malignancy in males worldwide—and therefore aids in monitoring public concerns about the health of men.

## Figures and Tables

**Figure 1 biosensors-12-00931-f001:**
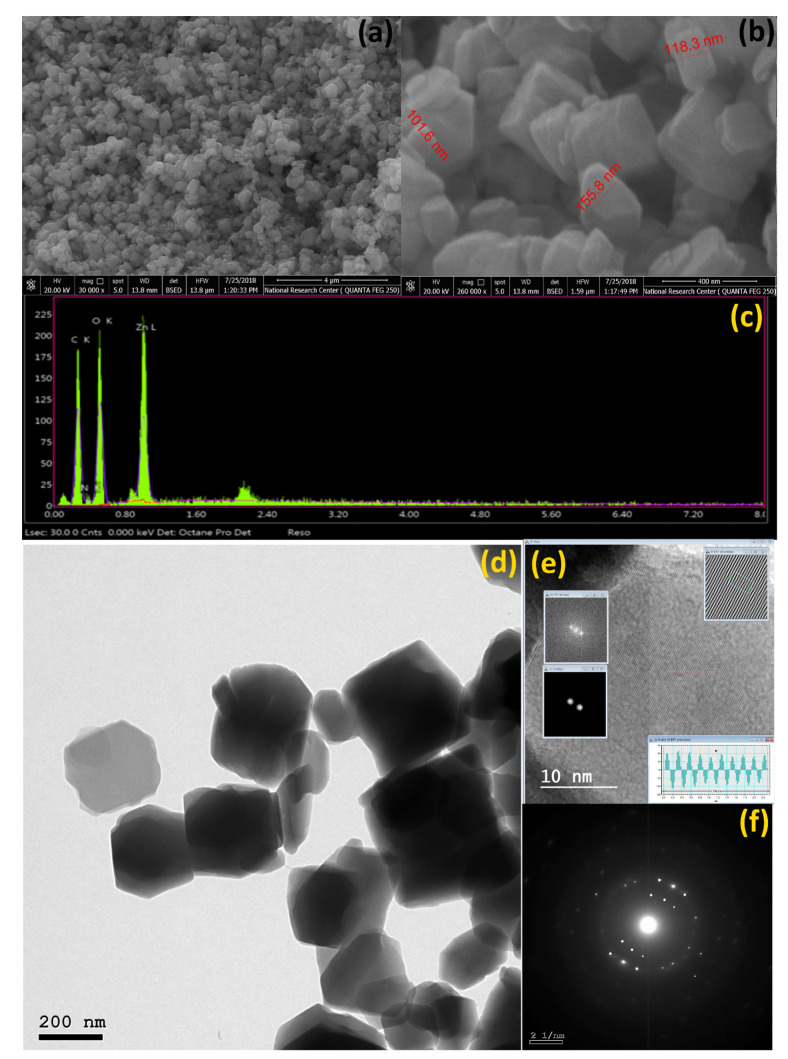
(**a**,**b**) FE-SEM images of the Zn-MOFs-NP at distinct magnifications, (**c**) mapping analysis (EDX) of the Zn-MOFs-NP, and (**d**–**f**) HR-TEM) and SAED images of the Zn-MOFs-NP.

**Figure 2 biosensors-12-00931-f002:**
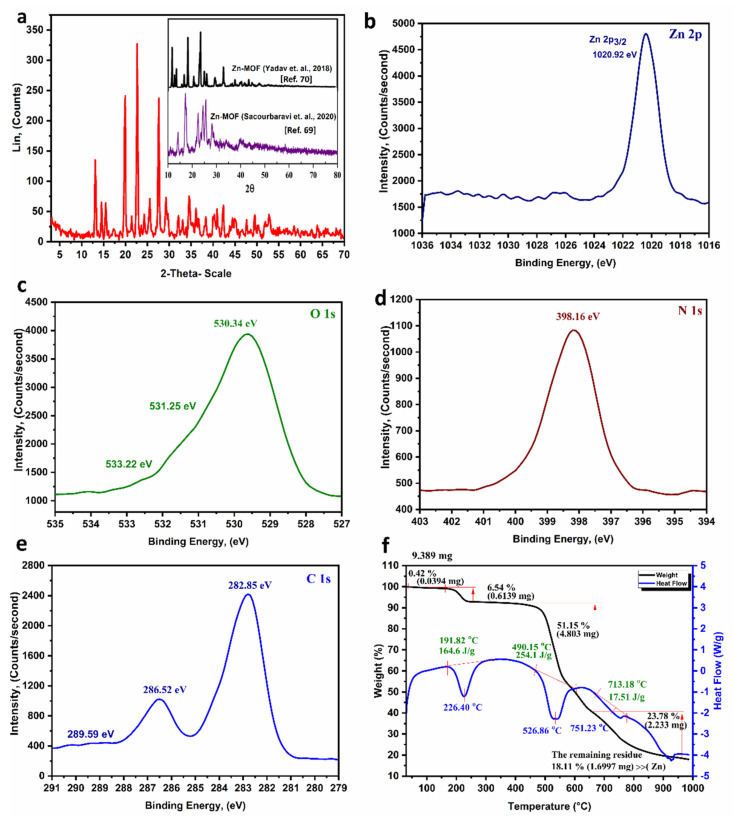
(**a**) The XRD patterns of the Zn-MOFs-NP, (**b**–**e**) the XPS spectra of the Zn-MOFs-NP: [Zn 2p (**b**), O 1s (**c**), N 1s (**d**), and C 1s (**e**)]; and (**f**) TGA-DSC of the Zn-MOFs-NP.

**Figure 3 biosensors-12-00931-f003:**
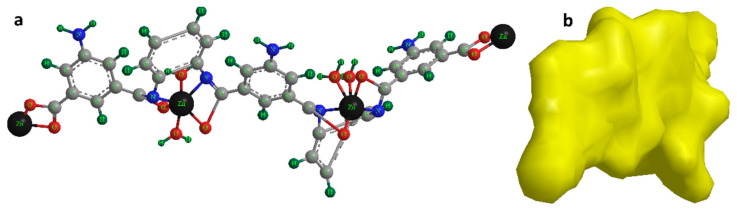
(**a**) The 3D suggested structural for the monomeric unit of the Zn-MOFs-NP; (**b**) Advanced molecular surface illustration.

**Figure 4 biosensors-12-00931-f004:**
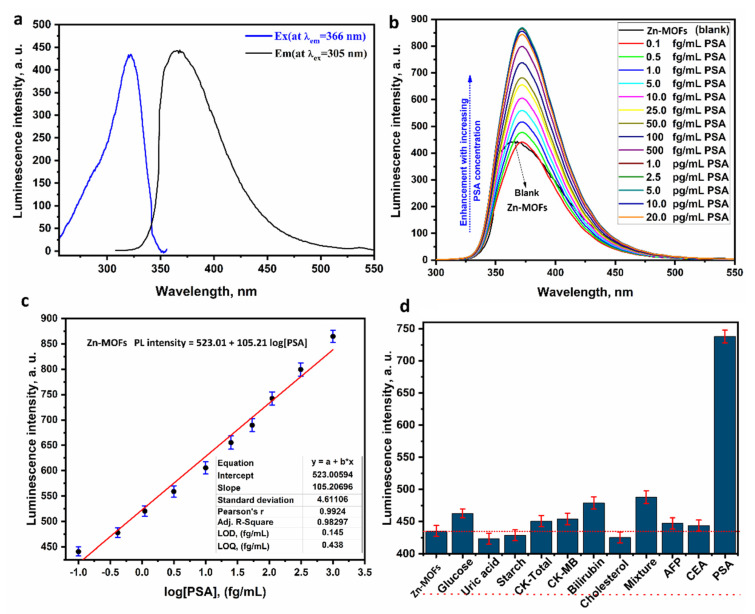
(**a**) The emission and excitation spectra (black and blue lines, respectively) of Zn-MOFs-NP, (**b**) the PL behavior of the Zn-MOFs-NP regarding various PSA concentrations, (**c**) a calibration graph between the PL intensities of the Zn-MOFs-NP and the logarithm of the PSA concentrations, and (**d**) a histogram of the PL intensities of the Zn-MOFs-NP regarding the PSA in the presence of some interfering analytes.

**Table 1 biosensors-12-00931-t001:** The elemental and EDX analysis data of the Zn-MOFs-NP compared with theoretically calculated.

Element	Theoretically Calculated	Found C/H/N Elemental Analysis	EDX Analysis (Found)			
			Weight%	Atomic%	Net Int.	Error%
Carbon	40.180	39.960	40.000	53.250	55.430	2.370
Hydrogen	5.760	5.950	-	-	-	-
Nitrogen	6.830	6.820	6.570	7.360	2.820	3.670
Oxygen	28.990	-	34.870	34.850	62.410	12.060
Zinc	18.230	-	18.560	4.540	30.60	1.310

**Table 2 biosensors-12-00931-t002:** The Zn-MOFs-NP biosensor in comparison to some existing methods for the determination of PSA.

Method	Linear Detection Range	LOD	Reference
Electrochemiluminescence	0.001–100.0 ng/mL	440 fg/mL	[4]
Nano Cu(II) complex biosensor	0.005–10,000 pg/mL	297 fg/mL	[10]
Impedimetric immunosensor	0.01–100 and 1–20,000 ng/mL	5.4 pg/mL	[15]
Voltammetry immunosensing platform	0.75–100.0 ng/mL	270 pg/mL	[8]
Chemiluminescence resonance energy transfer (CRET)	1.0–100 ng/mL	600 pg/mL	[16]
Electro chemiluminescent immunosensor	0.001–80 ng/mL	300 fg/mL	[26]
Colorimetric aptasensor	0.1–100 ng/mL	20.0 pg/mL	[18]
Fluorometric aptamer-based assay	0.05–150 pg/mL	43 fg/mL	[19]
Cooperate signal amplifications strategy	0.001–10,000 ng/mL	30 fg/mL	[22]
Photoelectrochemical immunosensor	0.02 pg/mL–200 ng/mL	6.8 fg/mL	[24]
Sandwich-type electrochemical immunosensor	1.0 pg/mL–100 ng/mL	0.45 pg/mL	[25]
Optical PL biosensor-based Zn-MOFs-NP	0.1 to 0.2 fg/mL	0.145 fg/mL	The present work

**Table 3 biosensors-12-00931-t003:** PSA determination in real serum samples using Zn-MOFs-NP biosensor.

Sample	Spiked PSA (fg/mL)	Found (fg/mL)			X	SD	RE %	R%
Serum samples	1.0000	0.9520	1.0280	0.9420	0.974	0.0470	1.0270	97.400
	100.00	101.13	96.90	95.200	97.74	3.0540	1.0340	96.740
	500.00	496.30	501.40	498.10	498.6	2.5870	1.0030	99.720
	1000.00	1007.7	1009.9	1018.0	1012	5.4240	0.9880	101.20
	2000.00	2002.6	2010.8	1992.0	2002.6	9.4250	0.9990	100.10

## Data Availability

All data generated or analyzed during this study are included in this published article (and its Appendix A).

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
