# Peer review of "Prostate-Specific Antigen Monitoring Using Nano Zinc(II) Metal–Organic Framework-Based Optical Biosensor"

_biosensors, 2022, doi:10.3390/bios12110931_

Round 1
Reviewer 1 Report
Comments to the authors: The authors of this manuscript developed a practical biosensor based on Zn(II) metal-organic framework nanoparticles (ZMOF-NPs). They used many spectroscopic and microanalytical tools to determine the properties of the prepared MOF. They found the ZMOF-NPs are sensitive to Prostate-Specific Antigen (PSA), selective to an extremely greater extent, and stable in terms of chemical composition. This work shows very realistic scientific significance. This topic is interesting and the results seem original. The approach appears technically sound and is presented clearly, I do have several concerns that should be addressed prior to my endorsement for publication. 1. In the introduction, it is necessary for the author to introduce the development of technologies and various schemes in this field. The specific performance parameters, advantages and disadvantages of various schemes need to be introduced in detail rather than simply mentioned. 2. The description of Figure 3 needs to be more detailed. For example, what does this 3D structure show? What is the auxiliary role for improving the sensing performance? 3. Please comment on the specific application scenarios of the synthesized nanostructures in the manuscript?Author Response
Reviewer 1:
Comments to the authors: The authors of this manuscript developed a practical biosensor based on Zn(II) metal-organic framework nanoparticles (ZMOF-NPs). They used many spectroscopic and microanalytical tools to determine the properties of the prepared MOF. They found the ZMOF-NPs are sensitive to Prostate-Specific Antigen (PSA), selective to an extremely greater extent, and stable in terms of chemical composition. This work shows very realistic scientific significance. This topic is interesting and the results seem original. The approach appears technically sound and is presented clearly, I do have several concerns that should be addressed prior to my endorsement for publication.
- Authors reply: We appreciate the time and efforts of the reviewer for this manuscript. Thank you so much for your comments and suggestions, that the reviewer’s comments are very critical points. Herein, we reply here point by point:
- In the introduction, it is necessary for the author to introduce the development of technologies and various schemes in this field. The specific performance parameters, advantages and disadvantages of various schemes need to be introduced in detail rather than simply mentioned.
- Author reply: Thank you very much for your valuable comment. Also, this point is raised also by the third respect reviewer. More clarifications were added in the revised version of the manuscript from line 51 to 69 with added supported references from number 28 to35.
- The description of Figure 3 needs to be more detailed. For example, what does this 3D structure show? What is the auxiliary role for improving the sensing performance?
- Author reply: Thank you very much for your valuable comment. More clarifications were added in the revised version of the manuscript. 3D structure of Zn-MOF plates and morphology have an intrinsic property such as photonic bandgap, microporosity, high-spatial density, low surface energy, the defects and edge positions, complexity over length scales ranging and integrated with functional low-dimensional building blocks which can enhanced the performance of sensing properties. As well as geometry with special crystal facets can provide a high large surface area and active sites, which is critically significant for highly efficient sensors
- Please comment on the specific application scenarios of the synthesized nanostructures in the manuscript?
- Authors reply: Thank you very much for your valuable comment. More clarifications related to specific application scenarios of the synthesized nanostructures were added in the revised version of the manuscript line 70 – 76 with supported references number 36-38.
- Finally, we are extremely grateful for the precious time, invaluable expertise, and superb professionalism you have put in improving the quality of our paper!

Reviewer 2 Report
The manuscript entitled “Prostate-specific antigen monitoring using nano zinc (II) metal-organic framework-based optical biosensor” reports the synthesis of practical bio-19 sensor based on Zn (II) metal-organic framework nanoparticles (ZMΟF-NPs). The authors characterized ZMΟF-NPs by Many spectroscopic and micro analytical tools which used to determine the structure, morphology, and physicochemical properties of the prepared MOF. The results indicates that, ZMΟF-NPs are sensitive to PSA, selective to an extremely greater extent, and stable in terms of chemical composition.
I have only minor comments for authors to address. My comments are:
1. The manuscript is very well written, but there are some typo errors in the manuscript. Authors must carefully check their submission. E.g., in the caption of Figure 4, the authors used ‘in present of’ instead of ‘in the presence of’. Also in section 2.5 (line number 9), represented ‘LD’ instead of ‘LOD’. In section 3 (line number 6), ‘building bulk’ instead of ‘building block’.
2. In section 2.4, PL-measurement procedure, the authors mentioned the concentration of stock solution after dissolving in DMSO is 1 mM. But they prepared 10 mM sample solution for their further studies which was synthesised by diluting the 1 mM stock solution using deionised water. Please check the sentence.
3. The peaks needs to be labelled in the FT-IR plot (Figure S4).
4. The experimental data (PL-measurement data) of the real sample analysis should be given in the supplementary informations.
5. The º C symbol should be uniform throughout the manuscript.
   
Author Response
Reviewer 2:
The manuscript entitled “Prostate-specific antigen monitoring using nano zinc (II) metal-organic framework-based optical biosensor” reports the synthesis of practical bio-19 sensor based on Zn (II) metal-organic framework nanoparticles (ZMΟF-NPs). The authors characterized ZMΟF-NPs by Many spectroscopic and micro analytical tools which used to determine the structure, morphology, and physicochemical properties of the prepared MOF. The results indicates that, ZMΟF-NPs are sensitive to PSA, selective to an extremely greater extent, and stable in terms of chemical composition.
I have only minor comments for authors to address. My comments are:
- Author reply: The authors appreciate the time and efforts of the reviewer for our manuscript. Thank you so much for your comments and suggestions which raise the quality of the present work to be suitable for publication in the “Biosensors” and to be meeting the journal requirements. Herein, we will reply here point by point for the comments:
- The manuscript is very well written, but there are some typo errors in the manuscript. Authors must carefully check their submission. E.g., in the caption of Figure 4, the authors used ‘in present of’ instead of ‘in the presence of’. Also in section 2.5 (line number 9), represented ‘LD’ instead of ‘LOD’. In section 3 (line number 6), ‘building bulk’ instead of ‘building block’.
- Author reply: Thank you very much for your valuable comment and your attention. You are completely right in your comment. Notably, the manuscript’s English was revised and improved seriously and re-checked for other mistakes in typing or grammar.
- In section 2.4, PL-measurement procedure, the authors mentioned the concentration of stock solution after dissolving in DMSO is 1 mM. But they prepared 10 mM sample solution for their further studies which was synthesised by diluting the 1 mM stock solution using deionised water. Please check the sentence.
- Author reply: Thank you very much for your valuable comment and your attention. Accordingly, more clarifications were added in section 2.4 in the revised version of the manuscript.
- The peaks needs to be labelled in the FT-IR plot (Figure S4).
- Author reply: Thank you very much for your valuable comment. The peaks labelled were added in the FT-IR plot in the revised version of the manuscript.
- The experimental data (PL-measurement data) of the real sample analysis should be given in the supplementary informations.
- Author reply: Thank you very much for your comment and your attention. You are completely right in your comment. This section is moved to PL-measurements procedures section in the revised version of the manuscript.
- The º C symbol should be uniform throughout the manuscript.
- Author reply: Thank you very much for your valuable comment and your attention. The symbol throughout the manuscript were carefully revised and rechecked and unified accordingly.
- Finally, we sincerely appreciate this respected reviewer’s very insightful, professional, and helpful comment, which indeed very significantly improve the quality of the work and the manuscript. Notably, we have tried our best to improve the work according to the valuable deep comments.

Reviewer 3 Report
Zn-MOF nanoparticles were used to develop a biochemical sensor and determine PSA with the LOD of 0.145 fg/mL in the range of 0.1-20 fg/mL. This manuscript is well organized and seems interesting. I recommend its possible publication in Biosensors after replying the following comments.
1. In the introduction, the PSA measurement methods or works should be reviewed and compared with the proposed biosensors to reveal your contribution or difference, especially for the PSA sensing performance based on other nanomaterials or the similar Zn/MOF materials.
2. Pay attention to the specification and unified expression of the physical units of temperature, see line 98 and line 105; wavelength, see line 117; etc.
3. In figure 2a and its sub-figures, the numbers in longitudinal coordinates has been removed, why?
4. In line 51, the authors said “the systems and procedures described above offer several benefits and drawbacks”, these properties should be discussed in detail. Because every sensor will not be perfect enough, even for the proposed on.
Author Response
Reviewer 3:
Zn-MOF nanoparticles were used to develop a biochemical sensor and determine PSA with the LOD of 0.145 fg/mL in the range of 0.1-20 fg/mL. This manuscript is well organized and seems interesting. I recommend its possible publication in Biosensors after replying the following comments.
- Author reply: The authors appreciate the time and efforts of the reviewer for our manuscript. Thank you so much for your comments and suggestions which raise the quality of the present work to be suitable for publication in the “Biosensors” and to be meeting the journal requirements. Herein, we will reply here point by point for the comments:
- In the introduction, the PSA measurement methods or works should be reviewed and compared with the proposed biosensors to reveal your contribution or difference, especially for the PSA sensing performance based on other nanomaterials or the similar Zn/MOF materials.
- Authors reply: We sincerely appreciate the respected reviewer for very insightful, professional, and helpful comment. Accordingly, more clarifications were added in the introduction section in the revised version of the manuscript line 70 – 76 with supported references number 36-38.
- Pay attention to the specification and unified expression of the physical units of temperature, see line 98 and line 105; wavelength, see line 117; etc.
- Author reply: Thank you very much for your valuable comment and your attention. The expression of the physical units of temperature and symbols throughout the manuscript were carefully revised and rechecked and unified accordingly.
- In figure 2a and its sub-figures, the numbers in longitudinal coordinates has been removed, why?
- Author reply: Thank you very much for your valuable comment and your attention. The numbers in longitudinal coordinates were added in the revised version of the manuscript.
- In line 51, the authors said “the systems and procedures described above offer several benefits and drawbacks”, these properties should be discussed in detail. Because every sensor will not be perfect enough, even for the proposed on.
- Authors reply: Thank you very much for your valuable, professionalism and deep comment. Also, this point is raised also by the first respect reviewer. Notably, for more clarifications, we mention that many methods and procedures used for early prostate cancer diagnosis as example like electrochemical and fluorometric methods in the references [14–26]. Each method and technique have advantages and limitations some of this method or technique is sensitive and accurate but are time consuming and solvent-usage intensive. However, the photoluminescence applications showed a distinctive advantages like very fast response, reusability, selectivity, sensitivity, operability, applicability, not require high-cost equipment. Accordingly, more clarifications were added in the introduction section in the revised version of the manuscript line 51 – 76 with supported references number 28-38.
- Finally, we sincerely appreciate this respected reviewer’s very insightful, professional, and helpful comment, which indeed very significantly improve the quality of the work and the manuscript. Notably, we have tried our best to improve the work according to the valuable deep comments.

Reviewer 4 Report
“Prostate-specific antigen monitoring using nano zinc (II) metal- 2 organic framework-based optical biosensor” is a manuscript by Said M. El-Sheikh et. al., for the detection of PSA biomarkers using Zn-MOF-NPs with 0.145 fg/ml LOD. While the concept is novel and interesting for the readers of Biosensors, the following issues should be considered and relative decisions made.
1. “ZMOF-NPs” is meaningless, it should be Zn-MOF. In the figures, it has been mentioned correctly, but in the whole text, it has been used wrongly.
2. The synthesized nano zinc(II) metal-organic framework belongs to which family of MOF? MOF-74, ZIF-69, MOF-5?
3. The claimed 0.145 fg/ml LOD is far better than any reported literature. The author should provide the calibration graph between the intensity of the Zn-MΟF-NPs and the PSA concentrations (not the logarithm of the PSA concentrations as presented in figure 4c).
4. Figure 1a resolution is too poor.
5. Section “2.3. Synthesis of ZMΟF-NPs”, cited reference “45” is not related to the synthesis of Zn-MOF. Cite the proper reference.
6. The Zn-MOF activation step has been skipped.
7. The data presented in table 2 regarding previous studies are wrong and different from the original literature. (LOD in reference 17 “https://doi.org/10.1016/j.ab.2018.10.024” is 0.6 ng/ml for PSA in the original literature while in table 2 it is 6.8 fg/ml. and in the same manner, reference 18 “https://doi.org/10.1016/j.bios.2013.08.042” is about “Environmental estrogens” detection and not PSA. However, the LOD for estradiol has changed from 0.015 pg/mL to 0.45 pg/mL).
8. Figures 1e and 1f have not been discussed in the manuscript.
9. Figure 4b, explain the reason for redshifting.
10. Figure 4d has not been introduced and discussed in the manuscript.
11. pH stability test and selectivity were not performed.
12. How did the author manage the concentration of PSA? I could not find any section regarding extraction or measurement of the PSA concentration.
13. Conclusion and abstract are very similar. Please rewrite the conclusion.
Author Response
Reviewer 4:
“Prostate-specific antigen monitoring using nano zinc (II) metal- 2 organic framework-based optical biosensor” is a manuscript by Said M. El-Sheikh et. al., for the detection of PSA biomarkers using Zn-MOF-NPs with 0.145 fg/ml LOD. While the concept is novel and interesting for the readers of Biosensors, the following issues should be considered and relative decisions made.
- Authors reply: We appreciate the time and efforts of the reviewer for this manuscript. Thank you so much for your comments and suggestions, that the reviewer’s comments are very critical points. Herein, we reply here point by point:
- “ZMOF-NPs” is meaningless, it should be Zn-MOF. In the figures, it has been mentioned correctly, but in the whole text, it has been used wrongly.
- Authors reply: Thank you very much for your valuable comment. You are partially right in your comment.
- . The abbreviations throughout the manuscript were carefully revised and modified accordingly.
- Referred to “ZMOF-NPs” is meaningless … wrongly”; in fact, the word "zinc metal organic framework" does not have a defined shorthand from IUPAC or any other high institution that is in charge of naming scientific acronyms. Accordingly, during our survey in the scientific publisher's websites you can see below some examples of many respectable manuscripts in high-ranked journals that used the same abbreviation.
- Some examples of manuscripts published in respectable journals used the ZMOF abbreviation for (Zinc-MOF):
(1) https://doi.org/10.1016/j.ijheatmasstransfer.2022.122667;
(2) https://doi.org/10.1016/j.inoche.2022.109436
(3) https://onlinelibrary.wiley.com/doi/full/10.1002/eem2.12320
- Referred to the abbreviations in the Figures for unity were carefully revised and modified accordingly.
- The synthesized nano zinc(II) metal-organic framework belongs to which family of MOF? MOF-74, ZIF-69, MOF-5?
- Authors reply: We sincerely appreciate the respected reviewer for very insightful, professional, and helpful comment. You highlighted a very critical point, the synthesized nano zinc(II) metal-organic framework is a novel MOF based on novel organic linker prepared by our group and belongs to a family of Salen-metal-organic framework (1-5)*.However, the diffraction patterns corresponded to normal Zn-MOF and other produced ZMΟF-NPs XRD patterns
*References:
- A.M. Shultz, O.K. Farha, D. Adhikari, A.A. Sarjeant, J.T. Hupp, S.T. Nguyen, Selective Surface and Near-Surface Modification of a Noncatenated, Catalytically Active Metal-Organic Framework Material Based on Mn(salen) Struts, Inorg. Chem. 50 (2011) 3174–3176.
- A.M. Shultz, A.A. Sarjeant, O.K. Farha, J.T. Hupp, S.T. Nguyen, Post-Synthesis Modification of a Metal À Organic Framework To Form, J. Am. Chem. Soc. 133 (2011) 13252–13255. https://doi.org/10.1021/ja204820d.
- A. Bhunia, Y. Lan, A.K. Powell, Salen-based metal–organic frameworks of nickel and the lanthanides, Chem. Commun. 47 (2011) 2035–2037. https://doi.org/10.1039/c0cc04881j.
- G. Salassa, M.J.J. Coenen, S.J. Wezenberg, B.L.M. Hendriksen, S. Speller, J.A.A.W. Elemans, A.W. Kleij, Extremely Strong Self-Assembly of a Bimetallic Salen Complex Visualized at the Single-Molecule Level, J. Am. Chem. Soc. 134 (2012) 7186−7192.
- L. Leoni, A.D. Cort, The Supramolecular Attitude of Metal – Salophen and Metal – Salen Complexes, Inorganics. 6 (2018) 42. https://doi.org/10.3390/inorganics6020042.
- The claimed 0.145 fg/ml LOD is far better than any reported literature. The author should provide the calibration graph between the intensity of the Zn-MΟF-NPs and the PSA concentrations (not the logarithm of the PSA concentrations as presented in figure 4c).
- Authors reply: We sincerely appreciate the respected reviewer for very insightful, professional, and helpful comment. You highlighted also a very critical point; we used the logarithm of the PSA concentrations due to the wide linear dynamic range from 0.1 fg/mL-20 pg/mL (0.0001- 20.0 pg/mL) the error of case using this range of concentrations without using the logarithm well be considered due to cut off values. As well as, we can not calculate the kinetic, slope and R2, standard deviation, LOD, and LOQ, with accuracy.
- Figure 1a resolution is too poor.
- Authors reply: We sincerely appreciate the respected reviewer for very insightful, professional, and helpful comment, which indeed very significantly improve the quality of the work and the manuscript. We understand the reviewer’s comment because we tried to collect many Figures in one object to reduce the publication area. However, as suggested, we have improved the quality of the Figure. If the figures still seem not good, it may be because of the PDF conversion process by the online submission system. Nevertheless, we will provide separate high-resolution figures and schemes during the production period.
- Section “2.3. Synthesis of ZMΟF-NPs”, cited reference “45” is not related to the synthesis of Zn-MOF. Cite the proper reference.
- Authors reply: Thank you very much for your comment. There is a conflict in this section. The cited reference was mention also in the introduction section and this reference is a cornerstone of the work which cited the previously report of nano organic linker synthesis. Accordingly, more clarification was added in this section and the reference number became 57.
- The Zn-MOF activation step has been skipped.
- Authors reply: We sincerely appreciate the respected reviewer for very insightful, professional, and helpful comment. You highlighted a very critical point, we agree with referee comment 100% that the activation step is very important in general for MOF and zeolites especially in sorption or adsorption of gases such occurring in BET measurements, the degassing process and defined temperature are key point in the measurements. In our case, It is less important due to we are working in the aqueous medium and the luminescence of Zn(II) MOFs mainly originates from the π–π* transition of ligand and on the other hand, this transition is mostly affected by guest molecules and solvent polarity. Although, we will care about this vital step in the next time.
- The data presented in table 2 regarding previous studies are wrong and different from the original literature. (LOD in reference 17 “https://doi.org/10.1016/j.ab.2018.10.024” is 0.6 ng/ml for PSA in the original literature while in table 2 it is 6.8 fg/ml. and in the same manner, reference 18 “https://doi.org/10.1016/j.bios.2013.08.042” is about “Environmental estrogens” detection and not PSA. However, the LOD for estradiol has changed from 0.015 pg/mL to 0.45 pg/mL).
- Authors reply: Thank you very much for your valuable comment and your attention. There are some mistyping and conflict come when the authors change the units (from ng/mL or pg/mL to fg/mL) for example, to be easy for the readers in case of making a comparison. As well as, a huge issue occurs when insert citation by Mendeley for ref. no. 18.
- For more clarification, in ref. no. 4 the authors mention the detection limit is as low as 0.44 pg mL−1, in our table we write LOD= 440 fg/mL to be easier for comparing in the current work. Accordingly, all the data in the presented in table 2 regarding previous studies were revised again from the original literature. As well as, all references were revised again.
- Figures 1e and 1f have not been discussed in the manuscript.
- Authors reply: We sincerely appreciate the respected reviewer for very insightful, professional, and helpful comment, which indeed very significantly improve the quality of the work and the manuscript. More clarifications were added in the revised version of the manuscript.
- Figure 4b, explain the reason for redshifting.
- Authors reply: We deeply appreciate the reviewer valuable deep comment for improving our work. Notably, the mechanism of the red-shift (about six nm) from 366.0 to 372.0 nm occurs because long-wavelength excitation results in photo selection of those fluorophores which are interacting most strongly with the polar solvent molecules. Moreover, MOF-containing Zn(II) nodes and conjugated aromatic linkers are well-known photoactive materials. Due to the d10 electron configuration of Zn2+, the luminescence behavior of these materials can be presumed to the π–π* transition of the ligand. Upon complex formation, a red shift in the emission spectrum of the MOF, with respect to that of its free ligand, is observed. Because Zn(II) ions are difficult to oxidize or reduce, the emissions of Zn(II) MOFs are neither metal-to-ligand charge transfer nor ligand-to-metal charge transfer. Thus, the emission is probably attributed to the intra-ligand π–π* transitions, perturbed by metal coordination. In the Zn(II) carboxylate-based MOFs, the highest occupied molecular orbital (HOMO) is likely the π-bonding orbital from the aromatic rings and the lowest unoccupied molecular orbital (LUMO) is related mainly to the Zn–O (carboxylate) π*-antibonding orbital, which is localized often on the metal centers. As the luminescence of Zn(II) MOFs mainly originates from the π–π* transition of ligand and on the other hand, this transition is mostly affected by guest molecules and solvent polarity changes; hence, potential luminescence properties of the Zn-MOF are first investigated in various solvents.
- Figure 4d has not been introduced and discussed in the manuscript.
- Authors reply: Thank you very much for your valuable comment. You are completely right in your comment, we agree with the respected reviewer. More clarifications were added in the revised version of the manuscript.
- pH stability test and selectivity were not performed.
- Authors reply: We sincerely appreciate the respected reviewer for very insightful, professional, and helpful comment, which indeed very significantly improve the quality of the work and the manuscript. More clarifications were added in the revised version of the manuscript. 4d histogram demonstrate this point, it can be noted that the luminescence intensity in case of PSA is highly enhanced, whereas, the other interfering analytes (similar biomarkers or organic molecules) doesn’t make significant changes in the luminescence intensity of ZMΟF-NPs-based biosensor. Which demonstrate the highly selectivity and specificity of the proposed biosensor. Additionally, various buffers (acetate, citrate, phosphate, and Britton-Robinson buffers) were investigated at the optimal pH, and phosphate buffer exhibits the best response among others. Finally, 0.1 mL phosphate buffer pH 6 was used to adjust the pH through the recommended procedure.
- How did the author manage the concentration of PSA? I could not find any section regarding extraction or measurement of the PSA concentration.
- Authors reply: We sincerely appreciate the respected reviewer for valuable comment. We mention this point in section 2.1. Materials, in which different buffered concentrations of standard-PSA ready to used were supplied by Monobind, USA.
- Conclusion and abstract are very similar. Please rewrite the conclusion.
- Authors reply: Thank you very much for your valuable comment. Actually, the conclusion and abstract are commentary to each other. The reptation in conclusion section was deleted and this section was revised again.
- Finally, we sincerely appreciate this respected reviewer’s very insightful, professional, and helpful comment, which indeed very significantly improve the quality of the work and the manuscript. Notably, we have tried our best to improve the work according to the valuable deep comments.

Round 2
Reviewer 1 Report
The current version of the manuscript is acceptable.
Author Response
Reviewer 1:
Comments to the authors: The current version of the manuscript is acceptable.
- Authors reply: We appreciate the time and efforts of the reviewer for this manuscript. Thank you so much for your valuable expertise, and superb professionalism comments and suggestions which raised the quality of the present work to be suitable for publication in the “Biosensors” and to be meeting the journal requirements. Moreover, all the cited references relevant to the research were revised again.

Reviewer 4 Report
1. ZMOFs is an abbreviation for “Zeolite-like metal−organic frameworks”. Even a simple search of “ZMOFs” in “google scholar” and “web of science” does not show significant results related to "zinc metal-organic framework". Therefore, to avoid confusion, the author must use Zn-MOFs as the abbreviation for "zinc metal-organic framework".
2. Figure 2a, the author has mentioned two references in the figure and abbreviated "zinc metal-organic framework" as Zn-MOFs.
3. Author has pointed out in line 129 that “N1, N3-bis(2-(3-((l1-oxidaneyl)carbonyl)-5-aminoben- zamido)phenyl)-5-aminoisophthalamide (ABAPAPA) linker which prepared according to previous report [57]”. As per my check in reference 57 “https://doi.org/10.1039/C9RA03030A” (which is published by same author in 2019) there is no report of preparation of the following linker.
4. The claimed 0.145 fg/ml LOD is far better than any reported literature. The author should provide the calibration graph between the intensity of the Zn-MΟF-NPs and the PSA concentrations (the logarithm of the PSA concentrations as presented in figure 4c is not acceptable).
5. Author should calculate the kinetic, slope and R2, standard deviation, LOD, and LOQ, with accuracy.
